# Effect of Thermal Modification Treatment on Some Physical and Mechanical Properties of *Pinus oocarpa* Wood

**Jhon F. Herrera-Builes** [1,*] **, Víctor Sepúlveda-Villarroel** [2] **, Jairo A. Osorio** [3] **, Linette Salvo-Sepúlveda** [2] **and Rubén A. Ananías** [2,*]

1 Departamento de Ciencias Forestales, Facultad de Ciencia Agrarias, Universidad Nacional de Colombia, Medellín 050034, Colombia

2 Research Group on Wood Drying & Heat Treatments, Wood Engineering Department, University of Bio-Bio, Concepción 4081112, Chile; vsepulveda@ubiobio.cl (V.S.-V.); lsalvo@ubiobio.cl (L.S.-S.)

3 Departamento de Ingeniería Agrícola y Alimentos, Facultad de Ciencias Agrarias, Universidad Nacional de Colombia, Medellín 050034, Colombia; aosorio@unal.edu.co

* Correspondence: jfherrer@unal.edu.co (J.F.H.-B.); ananias@ubiobio.cl (R.A.A.)

**Abstract:** This study deals with the effect of heat treatment on *Pinus oocarpa* specimens from forest plantations in Colombia. The effects of two heat treatments at 170 and 190 °C for 2.5 h in saturated vapor were evaluated based on the color, dimensional stability, air-dry and basic densities, modulus of elasticity (MOE), and modulus of rupture (MOR) in static bending of samples. The evaluations were carried out following the Colombian Technical Standards NTC 290 and 663, and the color changes resulting from heat treatments were monitored using the CIE-Lab, as well as other standards from the literature. The results show that there was 2.4% and 3.3% mass loss of wood modified at 170 and 190 °C, respectively. The air-dry and basic densities were higher in 170 °C treatment than after 190 °C treatment, and the thermal modifications applied increased the dimensional stability of the treated wood. After treatment at 170 and 190 °C, the lightness to darkness (L*) was reduced by 10% and 22%; the a* coordinate increased by 11% and 26%, causing redness in the treated wood; the b* coordinate increased by 14% and 17%; and the values of the wood color saturation (c*) increased by 14% and 18%, respectively. The general color change (ΔE*) increased gradually with the increase in the treatment temperature, resulting in a high color change to a very different color. The bending strength of thermally modified wood was improved and significantly increased to values higher than those of unmodified *Pinus oocarpa* wood. The high air-dry and basic densities, improved dimensional stability and resistance to bending, and attractive appearance of the treated wood indicate that thermal modification is a promising alternative for the transformation of *Pinus oocarpa* wood into a raw material with a high added value.

**Keywords:** dimensional stability; heat treatment; mechanical properties; thermally modified wood

## 1. Introduction

As a result of the increasing need to reduce environmental pollution, lumber producers around the world have gradually begun to reduce the amount of chemicals used to improve wood properties [1]. For this reason, thermal modification is an environmentally friendly alternative that can be applied to wood [2]. It is an efficient process in its use of resources, from which wood products with a lower environmental impact and extended service life are achieved, increasing their durability, stability, and properties [3]. The new characteristics of thermally modified wood make softwood species more attractive by drastically changing their properties [4]. In addition, the thermal modification of softwoods at temperatures below 200 °C minimizes the production of wood dust during remanufacture processes [5]. The heat-treatment process involves temperatures between 150 and 260 °C for times ranging from a few minutes to several hours, depending on the degree of modification desired [3,6]. Among the benefits of thermally modifying the wood, it was found that

the dimensional stability of the wood increases [7–9], presenting low values of shrinkage and swelling [8], owing to the changes in the polymeric components of the cell wall. This prevents the bonding between water molecules and the cellulose structure [4,9–12], which decreases the hygroscopic property of the treated wood [13]. Furthermore, the wood color change is an important parameter in evaluating the quality of the products obtained by their appearance. Therefore, the darkening of the wood obtained during the thermal modification by the degradation of hemicellulose and lignin—as well their conversion into extractive chromophores—provides improvements in the aesthetic properties of the treated wood [14–19].

Moreover, the mechanical properties can be affected depending on the intensity of the treatment. For this reason, it is important to identify the best conditions of the process for each species of wood and avoid a high reduction in mechanical properties, which could limit the range of applications of the material [20,21]; the modulus of rupture and modulus of elasticity in static bending are remarkably decreased after the heat treatment at temperatures above 200 °C [22–26].

According to studies reporting other wood species around the world, it is known that heat-treatment methods at 170 and 190 °C modify the physical and mechanical properties of plantation wood of *Pinus* species, which develop in different thermal levels from warm to cold [27]. Timber from plantations of fast-growing species has become an important resource for industry because of the decrease in forests of hardwood species and their acceptable values in terms of resistance, durability, and appearance [4,28]. However, *Pinus oocarpa* wood has low dimensional stability and low resistance mechanical properties, and it is very prone to being attacked by biological agents, limiting its use for structural purposes [26,28]. Considering the aforementioned characteristics, the thermal modification method could be an alternative to improve its properties and appearance, in turn reducing the demand for tropical wood [4]. Therefore, the objective of this study was to evaluate the effect of thermal modification treatments on the density, dimensional stability, color, static bending, and compression parallel to the grain. This research is expected to contribute to the knowledge about the short-rotation wood from *Pinus oocarpa* plantations to be used as alternatives in structural materials, therefore contributing significantly to the growth of the forest sector in Colombia.

## 2. Materials and Methods

### 2.1. Wood Specimens

Thirty 25-year-old *Pinus oocarpa* trees were collected, with a useful stem possessing a diameter at breast height of 45 cm. The wood was provided by the Empresa de Cipreses de Colombia SA and came from plantations located in the municipality of Yolombó (Antioquia, Republic of Colombia), from the farm La Carolina, lot Las Nubes, stand R2 (coordinates X = 891,818, Y = 1,227,491). For the air-dry density, basic density, and dimensional stability, 60 samples 30 mm × 30 mm × 100 mm (tangential × radial × longitudinal) were used. For the mechanical properties, 42 samples 25 mm × 25 mm × 410 mm (tangential × radial × longitudinal) were used for the static bending. The samples used for these tests were free of knots and defects, with well-directed growth rings and clear radial and tangential orientations. After thermal treatment, the samples were kept in a climate chamber (Binder KMF-115, Binder, Tuttlingen, Germany) at constant conditions (20 °C, 65% RH) and an equilibrium moisture content (EMC) of 12%.

### 2.2. Thermal Modification

The thermal modification of *Pinus oocarpa* specimens at 170 and 190 °C was carried out in a prototype chamber (Model Lab3.5e, Neumann, Concepción, Chile), with a fluid flow speed of 6 m/s. For each load of treatment, the chamber was loaded with 120 boards of 50 mm wood, which were stacked evenly and placed in layers of equal distance, with 20-mm-thick spacers to allow steam to move through the pile and distribute the weight vertically from top to bottom. The *Pinus oocarpa* specimens were stacked inside the middle

of the wood pile so that they were continuous and without empty spaces so as to prevent damage (Figure 1). The prototype chamber operated during the modification steps was under an atmosphere of steam, with a continuous flow of current without pressure. The dry and wet bulb temperatures, as well as the wood temperature (Figures 2 and 3), were monitored according to the configuration and the computerized kiln program (Canelo, Control Total, v8.0, Concepción, Chile). The thermal modification process was adapted from the literature [29,30] and began with a temperature increase rate of 1 °C/min until reaching 100 °C. The temperature was then maintained for 15–22 h, allowing the wood to dry 4–6%. Subsequently, a temperature increase rate of 0.7 °C/min was applied to reach either 170 or 190 °C. In this step, a steam atmosphere was used to reduce air and prevent damage. The treatment temperature (170–190 °C dry bulb or 100 °C wet bulb) was maintained for approximately 2.5 h. The last stage was the cooling and stabilization of the samples for approximately 5–7 h. The temperature gradient between the surface and the center of the samples did not exceed 15–20 °C, in order to maintain the quality of the specimens. The total work cycle was approximately 30 h, and the final moisture content of the specimens was 9–11%. Finally, *Pinus oocarpa* specimens treated at 170 and 190 °C were obtained and compared with the unmodified material.

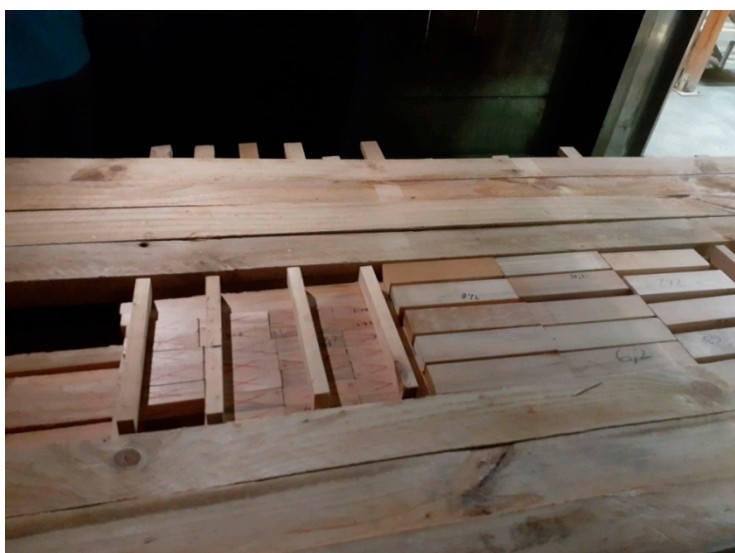

**Figure 1.** Stacking of *Pinus oocarpa* specimens within the drying pile.

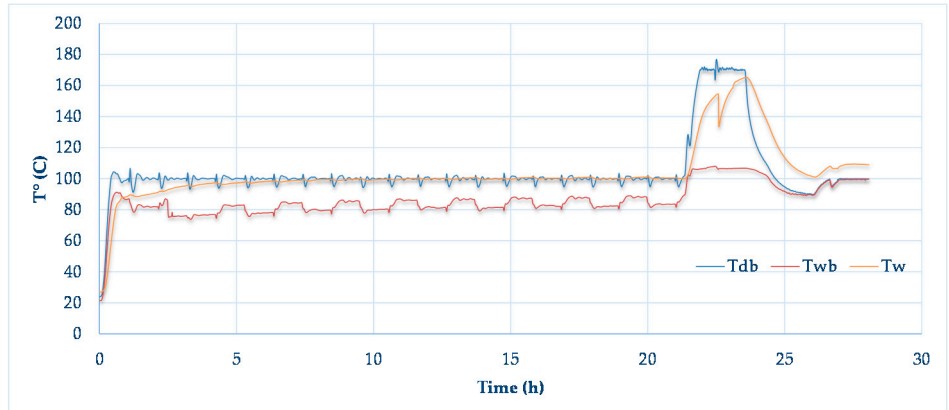

**Figure 2.** Evolution time of temperatures during the thermal modification process at 170 °C. Tdb—dry-bulb temperature; Twb—wet-bulb temperature; Tw—wood temperature.

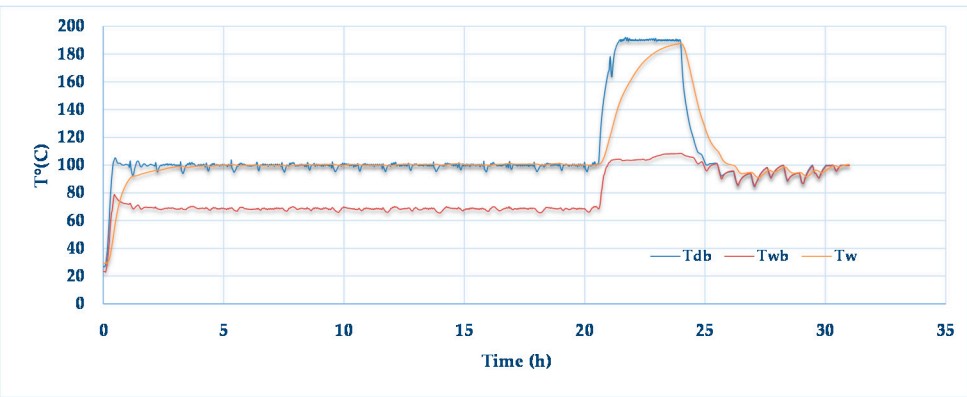

**Figure 3.** Evolution time of the temperatures during the thermal modification process at 190 °C. Tdb—dry-bulb temperature; Twb—wet-bulb temperature; Tw—wood temperature.

### 2.3. Physical Characterization

Both thermally modified and unmodified *Pinus oocarpa* specimens were subjected to density tests, following the guidelines of the Colombian Technical Standard NTC 290 [31]. Loss of mass (ML) was determined by measuring the difference between the initial mass and the mass after each treatment, following the calculation method applied by Kamperidou [2]. Dimensional stability was assessed by the Rowell and Youngs test [32], in a complete cycle, which consisted of passing the wood from dry to wet and then wet to dry successively until three cycles were completed. The drying was carried out in an oven at 103 ± 2 °C until reaching a constant weight, and the wet cycle was carried out by soaking the samples for 24 h and then measuring the change in dimensions of the samples. The volumetric swelling coefficient (S) and the anti-swelling efficiency (ASE) were estimated considering three cycles as a measure of the dimensional stability of the wood. The following equations were used to calculate the volumetric swelling coefficient and the anti-swelling efficiency, respectively:

$$S\% = 100(V_2 - V_1)/V_1, \tag{1}$$

where $V_2$ is the volume of the saturated samples and $V_1$ is the volume of the oven-dried samples.

$$ASE\% = 100(S_s - S_m)/S_s, \tag{2}$$

where $S_s$ is the average coefficient of the unmodified specimens and $S_m$ is the average coefficient of the thermally modified specimens.

### 2.4. Color

Color changes as a result of heat treatments were monitored using the CIE-Lab color space coordinate system. Here, 28 samples per treatment were measured at different points on the surface (three measurements for each sample), according to ASTM D2244–09b [33], using a spectrophotometer device (Datacolor check 3, LAV/USAV model, Datacolor, NJ, USA), and the results were expressed according to Equations (3) and (4):

$$\Delta E^* = \sqrt{\Delta L^{*2} + \Delta a^{*2} + \Delta b*^2}, \tag{3}$$

where $\Delta E^*$ is the total color difference between the lightness ($\Delta L^*$), the red–green axis ($\Delta a^*$), and the yellow–blue axis ($\Delta b^*$) before and after the modification. The evaluation criteria for general color changes were chosen based on the work of Barcík et al. [34] and are as follows: 0.2 < $\Delta E^*$ (invisible difference), 0.2 < $\Delta E^*$ < 2 (small difference), 2 < $\Delta E^*$ < 3 (visible color change with high quality filter), 3 < $\Delta E^*$ < 6 (visible color change with medium quality filter), 6 < $\Delta E^*$ < 12 (high color changes), and $\Delta E^*$ > 12 (different color). In addition,

the color saturation with chroma (*C\**) was determined based on the CIE-Lab parameters, according to Equation (4):

$$C^* = \sqrt{a^{*2} + b^{*2}}. \tag{4}$$

### 2.5. Mechanical Characterization

For the mechanical tests, the thermally modified and unmodified *Pinus oocarpa* specimens were subjected to static bending tests following the guidelines of the Colombian Technical Standard NTC 663 [35]. The determination of the mechanical properties of the untreated wood was carried out in the facilities of the Forest Products Laboratory of the Universidad Nacional de Colombia, Medellin, in a Tinius Olsen universal testing machine (Willow Grove, PA, USA) with a capacity of 147 kN. The mechanical properties tests of the thermally modified specimens were developed at the University of Bío-Bío, Chile, in an Instron universal testing machine with a capacity of 10 kN (São José dos Pinhais, Brazil).

### 2.6. Data Analysis

A completely randomized experimental design was established, with 3 treatments, 20 repetitions per treatment for the physical tests, and 14 repetitions per treatment for the mechanical tests. Analysis of variance (ANOVA) was performed for each variable, and when the F test showed significant results, a Tukey multiple range test ($p \leq 0.05$) was performed using the Statgraphics Centurion XVI statistical program. The information was processed based on the methodology described by Gómez [36].

## 3. Results

### 3.1. Physical Properties

Table 1 shows that the wood densities are affected by the temperature of the heat treatment. At 170 °C, the air-dry density increased by 5%, but at 190 °C, it decreased by 10%, presenting statistically significant differences for both heat treatments. For the basic density, on the other hand, the samples modified at 170 °C increased by 20%, but the wood treated at 190 °C did not improve, obtaining values similar to those obtained in the unmodified wood, which also presented statistically significant differences. These results are in disagreement with other studies in which the density decreased after thermal modification treatments [23,37,38]. This increase in the density, particularly at 170 °C, is probably due to the crosslinking of the wood caused by the polycondensation reactions of the lignin, as well as increases in molecular weight [39]. The mass loss increased with temperature. The losses that were obtained were 2.4% for the modification at 170 °C and 3.3% for the treatment at 190 °C, but this was not statistically significant. The mass loss values were similar to the results found in other studies on thermally modified softwoods [6,15,38].

**Table 1.** Wood densities and mass loss of thermally modified *Pinus oocarpa*.

| Density | Unmodified | 170 °C | 190 °C |
|---|---|---|---|
| Air-dry density (kg/m$^3$) | 630.47 ± 57.67 ab | 662.59 ± 46.10 a | 569.57 ± 43.67 b |
| Basic density (kg/m$^3$) | 521.30 ± 47.93 a | 627.12 ± 44.26 b | 524.29 ± 42.73 a |
| Mass loss (%) | - | 2.39 ± 0.17 a | 3.34 ± 0.34 a |

Note: The body of the table shows the average values ± a 95% confidence interval of 20 test samples per treatment. Equal letters in columns indicate no significant difference between treatments, according to Tukey's multiple range test ($p \leq 0.05$).

In Table 2, the anti-swelling efficiency (ASE) values of thermally modified *Pinus oocarpa* are presented, the ASE values obtained indicate that the treatment temperature had a significant effect on the improvement of the dimensional stability of the wood. In the radial orientation, the ASE values were 34.41% and 48.34% at 170 and at 190 °C, respectively, and in the tangential orientation, the ASE values were 42.69% and 54.12% at

170 and 190 °C, respectively. Both orientations showed statistically significant differences between the modification temperatures. These results were similar to the ASE values obtained for other thermally modified softwoods [6,9]. The ASE improvements were due to transformations of the chemical structure of wood according to the treatment temperature used, reducing the amount of hydroxyl groups caused mainly by the autocatalytic reactions of the constituents of the cell wall [8,12].

**Table 2.** Anti-swelling efficiency (ASE) of thermally modified *Pinus oocarpa*.

| ASE | 170 °C | 190 °C |
|---|---|---|
| Radial (%) | $34.41 \pm 0.71$ a | $48.34 \pm 1.56$ b |
| Tangential (%) | $42.69 \pm 0.70$ a | $54.12 \pm 1.62$ b |

Note: The body of the table shows the average values ± a 95% confidence interval of 20 test samples per treatment. Equal letters in columns indicate no significant difference between treatments, according to Tukey's multiple range test ($p \leq 0.05$).

The color parameters are reported in Table 3. Lightness to darkness (*L**) at a higher temperature decreased by 10% and 22% at 170 and 190 °C, respectively, showing statistically significant differences between treatments. These results are comparable to those obtained for other thermally modified pine woods [16,30]. The increase in the values of the *a** coordinate led to redness in the treated wood; the *a** coordinate increased by about 11% after the treatment at 170 °C and 26% after treatment at 190 °C, presenting statistically significant differences between the treatments. These results are comparable to those obtained in other conifers [16,30]. In relation to the increase in the *b** coordinate, which is associated with the yellowing of the treated wood, there were increases of 14% after the application of 170 °C and 17% after the application 190 °C, without statistically significant differences. The results obtained here are higher than those in the other reported studies [16,30,40]. The values of the wood color saturation (*c**) increased by 14% after the application of the 170 °C treatment and 18% after treatment at 190 °C, without statistically significant differences between treatments. The results of *c** were higher than those reported for other pinewood species [15,30,41]. The general color change (Δ*E**) increased gradually with the increase in the treatment temperature. After the application of 170 °C, a high color change was obtained (9.13%), and after treatment at 190 °C, the resulting color was very different (13.4%), which led to statistically significant differences. On the one hand, the reported color changes are probably related to the intensity of the treatment, and on the one hand, they might be related to the formation of chromogenic products due to the thermal degradation of some cell wall components, which are mainly connected to the lignin present in the wood [42].

**Table 3.** Color parameters *L**, *a**, *b**, *c**, and Δ*E** of thermally modified *Pinus oocarpa*.

| Parameters | Unmodified | 170 °C | 190 °C |
|---|---|---|---|
| *L** | $69.95 \pm 1.13$ a | $62.98 \pm 1.57$ b | $57.30 \pm 1.77$ c |
| *a** | $8.53 \pm 0.52$ a | $9.50 \pm 0.63$ b | $10.73 \pm 0.49$ c |
| *b** | $27.33 \pm 0.65$ a | $31.28 \pm 1.01$ b | $31.98 \pm 0.55$ b |
| *c** | $28.65 \pm 0.76$ a | $32.70 \pm 1.14$ b | $33.75 \pm 0.59$ b |
| Δ*E** | - | $9.13 \pm 1.85$ a | $13.49 \pm 2.14$ b |

Note: The body of the table shows the average values ± a 95% confidence interval of 20 test samples per treatment. Equal letters in columns indicate no significant difference between treatments, according to Tukey's multiple range test ($p \leq 0.05$).

### 3.2. Mechanical Properties

The static bending values obtained for the thermally modified *Pinus oocarpa* wood are presented in Table 4. The modulus of rupture (MOR) increased by 47% and 22% after the thermal modification treatment at 170 and 190 °C, respectively, presenting statistically

significant differences between the treatments. The thermal modifications reported contrast with the results reported for similar heat-treatment applications, in which reductions in MOR flexural strength greater than 30% are evidenced [35,43]. In the studied wood samples, the static flexural modulus of elasticity (MOE) increased by 4% and 10% after treatment at 170 and 190 °C, respectively; these results are in line with other studies [19,30].

**Table 4.** Static bending of thermally modified *Pinus oocarpa*.

| Mechanical Properties | Unmodified | 170 °C | 190 °C |
|---|---|---|---|
| MOR (MPa) | 83.88 ± 9.17 a | 123.67 ± 25.61 b | 102.76 ± 23.58 ab |
| MOE (GPa) | 11.37 ± 1.63 a | 11.83 ± 1.73 a | 10.18 ± 1.94 a |

Note: The body of the table shows the average values ± a 95% confidence interval of 20 test samples per treatment. Equal letters in columns indicate no significant difference between treatments, according to Tukey's multiple range test ($p \leq 0.05$).

The increase in mechanical properties is probably due to the crosslinking of the lignin network, as well as the rearrangement and crystallization of cellulose, acting as a hardener for the microfibrils and the middle lamina [44]. In particular, these results favor the application of *Pinus oocarpa* species for structural purposes, as the values obtained exceed the requirements established by the Colombian Standards for category C of lower-resistance woods (MOE > 11.5 and basic density > 560 kg/m$^3$), and thus the thermal modification treatments at 170 °C allowed for changing the classification of the *Pinus oocarpa* wood from category C to category B; this change to a better quality for *Pinus oocarpa* wood is due to the superior resistance obtained [45]. *Pinus oocarpa* thermally modified at 170 °C would produce higher wood quality than unmodified *Pinus oocarpa* wood. These experimental data obtained for wood sample dimensions are needed for validation of the results for thermal modification of *Pinus oocarpa* with sawn wood dimensions.

## 4. Conclusions

The effects of thermal modification treatment on *Pinus oocarpa* specimens were evaluated based on physical and mechanical properties.

The thermal modifications applied increased the dimensional stability of the treated *Pinus oocarpa* wood.

Thermal modification of *Pinus oocarpa* at 170 °C produced wood of higher quality than unmodified *Pinus oocarpa* wood.

Low mass loss of 2.4% and 3.3% of thermally modified pine wood for temperatures of 170 and 190 °C, respectively, is an important quality characteristic of the process applied in *Pinus oocarpa*.

The bending strength of thermally modified wood was improved and significantly increased to values higher than those of unmodified *Pinus oocarpa* wood.

The thermal modification changed the color parameters of the untreated wood; the principal changes were the increased saturation and darkening, reddening, and yellowing effects.

The experimental results indicated that the thermal modification of *Pinus oocarpa* wood at 170 °C yields wood of better quality by obtaining a higher density and resistance, darkened colors (without adding color), and improved dimensional stability that favors their application for structural purposes, in accordance with the requirements of the Colombian Standards.

**Author Contributions:** Conceptualization, J.F.H.-B.; methodology, J.F.H.-B., J.A.O., L.S.-S., and V.S.-V.; formal analysis, J.F.H.-B.; investigation, J.F.H.-B.; writing—original draft preparation, J.F.H.-B.; writing—review and editing, J.F.H.-B., R.A.A., J.A.O., L.S.-S., and V.S.-V.; supervision, R.A.A. and J.A.O. All authors have read and agreed to the published version of the manuscript.

**Funding:** This research received no external funding.

**Institutional Review Board Statement:** Not applicable.

**Informed Consent Statement:** Not applicable.

**Data Availability Statement:** The data presented in this study are available on request from the corresponding author.

**Acknowledgments:** The authors wish to thank the Company of Cipreses de Colombia S.A. for the donation of the wood material and the staff at the Forest Products Laboratory "Héctor Anaya López" of the Universidad Nacional de Colombia for their support and collaboration. The authors are also grateful for the support of the National Agency of Science and Development of Chile (Fondequip EQM130236).

**Conflicts of Interest:** The authors declare no conflict of interest.

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
