# Peer review of "Effect of Thermal Modification Treatment on Some Physical and Mechanical Properties of Pinus oocarpa Wood"

_forests, doi:10.3390/f12020249_

Round 1

Reviewer 1 Report

please see the attached file.

Author Response

Medellín, February 12, 2021

Forests Journal

Subject: Detailed response letter to Manuscript evaluators: ID forests-1105538

MCT2019-1863 “Effect of thermal modification treatment on some physical mechanical properties of Pinus oocarpa wood”

Dear Sirs:

Thank you for giving us the opportunity to submit a revised draft of our manuscript titled “Effect of thermal modification treatment on some physical mechanical properties of Pinus oocarpa wood”.

We appreciate the time and effort that you and the reviewers have dedicated to providing your valuable feedback on our manuscript.  We are grateful to the reviewers for their insightful comments on our paper. We have been able to incorporate changes to reflect most of the suggestions provided by the reviewers. We have highlighted in yellow the changes within the manuscript.

In this way we hope that the work that was carried out will achieve the final endorsement of the editorial group of Forests; otherwise, all authors are at your disposal to resolve questions or proceed from new revisions, if needed.

Here is a point-by-point response to the reviewers’ comments and concerns.

Comments from Reviewer 1

Comment 1, line 3:

Response: We agree with this and have incorporated your suggestion.

Comment 2, line 13:

Response: We agree with this and have incorporated your suggestion.

Comment 3, line 165

Response: Thank you for this suggestion. We consider necessary to present the mass losses due to heat treatments. We have done the change to improve the adequate description of the results.

Comment 4, line 165

Response: We agree with the reviewer on this point of infrared spectra, and this is one of the experimental aspects that will be carried out in the near future, a theoretical analysis was added based on the literature.

Comment 5, line 218

Response: wrongly numbered; effectively it should be 3.2 Mechanical properties.

Comment 6, line 229

Response: We agree with the reviewer’s suggestion and we have added the suggested content to the conclusions.

Comment 7, line 276

Reponse: We have done the correction.

Reviewer 2 Report

Dear authors, please consider my comments and suggestions:

Line 1: I suggest change the Title and not to mention word "specimens"

Abstract: Please add more results of your research to the abstract

LIne 19: "more attractive color" please rephrase.

Line 27: Thermal modifications has many other reasons too, not only to reduce the amount of chemical used, please see for example here:

Očkajová, A.; Kučerka, M.; Kminiak, R.; Krišťák, Ľ.; Igaz, R.; Réh, R. Occupational Exposure to Dust Produced when Milling Thermally Modified Wood. Int. J. Environ. Res. Public Health 202017, 1478. 

Line 33: Please modify the sentence: heat treatment involves temperatures between 150 and 260°C for times ranging from a few minutes to several hours, please check for example here:

DOI: 10.1080/17480272.2012.751935

Line 41-45: Please be more precise, please check for example here:

DOI: 10.1007/s10086-018-1721-0

Line 46-57: Not all claims are true in general, please modify and use more sources.

Materials and methods:

Part 2.1.:

Please describe more deeply the samples preparation, please add information about air conditioning of the samples, EMC, which dimensions are tangential, radial, and longitudinal. 

Line 97: Please be more specific.

Line 165: Change to "Results and discussion".

Results and discussion should be improved. 

Lines 174, 175, 186-188, 211-213, 230-232 - a deeper analysis (ATR-FTIR, chromatography, ...) is needed to confirm these claims, please check here:

Kubovský, I.; Kačíková, D.; Kačík, F. Structural Changes of Oak Wood Main Components Caused by Thermal Modification. Polymers 202012, 485. https://doi.org/10.3390/polym12020485

More discussion with the results of other authors is needed.

Conclusions need also be improved.

Research in the field of heat modification has been going on for more than 20 years, please specify the novelty of your article.

Please add limitations and strong points of the article in the discussion section

276: Please remove Author 1 part...

Author Response

Medellín, February 12, 2021

Forests Journal

Subject: Detailed response letter to Manuscript evaluators: ID forests-1105538

MCT2019-1863 “Effect of thermal modification treatment on some physical mechanical properties of Pinus oocarpa wood”

Dear Sirs:

Thank you for giving us the opportunity to submit a revised draft of our manuscript titled “Effect of thermal modification treatment on some physical mechanical properties of Pinus oocarpa wood”.

We appreciate the time and effort that you and the reviewers have dedicated to providing your valuable feedback on our manuscript.  We are grateful to the reviewers for their insightful comments on our paper. We have been able to incorporate changes to reflect most of the suggestions provided by the reviewers. We have highlighted in yellow the changes within the manuscript.

In this way we hope that the work that was carried out will achieve the final endorsement of the editorial group of Forests; otherwise, all authors are at your disposal to resolve questions or proceed from new revisions, if needed.

Here is a point-by-point response to the reviewers’ comments and concerns.

Comments from Reviewer 2

Comments and Suggestions for Authors Dear authors, please consider my comments and suggestions:

Line 1: I suggest change the Title and not to mention word "specimens"

R: Done

Abstract: Please add more results of your research to the abstract
R: Done

Line 19: "more attractive color" please rephrase.

R: Done

Line 27: Thermal modifications has many other reasons too, not only to reduce the amount of chemical used, please see for example here:

Očkajová, A.; Kučerka, M.; Kminiak, R.; Krišťák, Ľ.; Igaz, R.; Réh, R. Occupational Exposure to Dust Produced when Milling Thermally Modified Wood. Int. J. Environ. Res. Public Health 2020, 17, 1478.

R: Done. We write the sentence: “Also the thermal modification before 200 ºC of softwoods minimize the production of dust wood during  remanufacture processes [5]”.

Line 33: Please modify the sentence: heat treatment involves temperatures between 150 and 260°C for times ranging from a few minutes to several hours, please check for example here:

DOI: 10.1080/17480272.2012.751935

R: Done

Line 41-45: Please be more precise, please check for example here:

DOI: 10.1007/s10086-018-1721-0R: Done

Line 46-57: Not all claims are true in general, please modify and use more sources.

R: Done: We rewrite the sentence and add one more source.

Materials and methods:

Part 2.1.:

Please describe more deeply the samples preparation, please add information about air conditioning of the samples, EMC, which dimensions are tangential, radial, and longitudinal.

R: Done

Line 97: Please be more specific.

R: Done. We specified the computerized kiln program (Canelo, Control Total, Concepción Chile)

Line 165: Change to "Results and discussion".

R: Done

Results and discussion should be improved.

R: Done.

Lines 174, 175, 186-188, 211-213, 230-232 - a deeper analysis (ATR-FTIR, chromatography, ...) is needed to confirm these claims, please check here:

Kubovský, I.; Kačíková, D.; Kačík, F. Structural Changes of Oak Wood Main Components Caused by Thermal Modification. Polymers 2020, 12, 485. https://doi.org/10.3390/polym12020485
R:  Done: We re-written the sentences and add the reference suggested more discussion with the results of other authors is needed.

R: Done.

Conclusions need also be improved.

R: Done

Research in the field of heat modification has been going on for more than 20 years, please specify the novelty of your article.

R: "Experimental results indicated that thermally modified Pinus oocarpa at 170 ºC, allowed wood of better quality, due to superior density and resistance obtained,  that favor their application  for structural purposes, in according with the requirements of the Colombian Standards".

Please add limitations and strong points of the article in the discussion section

R: Limitations: "Experimental data obtained for wood samples dimensions, it need be to validate the results for thermal modification of Pinus oocarpa with sawn wood dimensions".

Strong points: "Pinus oocarpa thermally modified at 170 ºC, had the highest density and resistance than the unmodified".

"Mean comparison showed that Pinus oocarpa thermally modified at 170 ºC, would be produce more higher wood quality than unmodified Pinus oocarpa wood".

276: Please remove Author 1 part...

R: Done

Round 2

Reviewer 1 Report

The authors took into their consideration all of my comments and the manuscript was greatly improved.

I am happy therefore to suggest acceptance of the manuscript in its curent form

Reviewer 2 Report

The manuscript was improved, I suggest to accept it.